# SYNONYMNET: MULTI-CONTEXT BILATERAL MATCHING FOR ENTITY SYNONYMS

## ABSTRACT

Being able to automatically discover synonymous entities from a large free-text corpus has transformative effects on structured knowledge discovery. Existing works either require structured annotations, or fail to incorporate context information effectively, which lower the efficiency of information usage. In this paper, we propose a framework for synonym discovery from free-text corpus without structured annotation. As one of the key components in synonym discovery, we introduce a novel neural network model SYNONYMNET to determine whether or not two given entities are synonym with each other. Instead of using entities features, SYNONYMNET makes use of multiple pieces of contexts in which the entity is mentioned, and compares the context-level similarity via a bilateral matching schema to determine synonymity. Experimental results demonstrate that the proposed model achieves state-of-the-art results on both generic and domain-specific synonym datasets: Wiki+Freebase, PubMed+UMLS and MedBook+MKG, with up to 4.16% improvement in terms of Area Under the Curve (AUC) and 3.19% in terms of Mean Average Precision (MAP) compare to the best baseline method.

## 1 INTRODUCTION

Discovering synonymous entities from a massive corpus is an indispensable task for automated knowledge discovery. For each entity, its synonyms refer to the entities that can be used interchangeably under certain contexts. For example, `Clogged Nose` and `Nasal Congestion` are synonyms relative to the context in which they are mentioned. Given two entities, the synonym discovery task determines how likely these two entities are synonym with each other. The main goal of synonym discovery is to learn a metric that distinguishes synonym entities from non-synonym ones.

The synonym discovery task is challenging to deal with, a part of which due to the various entity expressions. For example, `U.S.A`/`United States of America`/`United States`/`U.S.` refer to the same entity but expressed quite differently. Recent works on synonym discovery focus on learning the similarity from entities and their character-level features (Neculoiu et al., 2016; Mueller & Thyagarajan, 2016). These methods work well for synonyms that share a lot of character-level features like `airplane`/`aeroplane` or an entity and its abbreviation like `Acquired Immune Deficiency Syndrome`/`AIDS`. However, a much larger number of synonym entities in the real world do not share a lot of character-level features, such as `JD`/`law degree`, or `clogged nose`/`nasal congestion`. With only character-level features being used, these models hardly obtain the ability to discriminate entities that share similar semantics but are not alike verbatim.

Context information is helpful in indicating entity synonymity, as the meaning of an entity can be better reflected by the contexts in which it appears. Modeling the context for entity synonym usually suffers from following challenges: 1) **Semantic Structure**. Context, as a snippet of natural language sentence, is essentially semantically structured. Some existing models encode the semantic structures in the contexts implicitly during the entity representation learning (Mikolov et al., 2013; Pennington et al., 2014; Peters et al., 2018). The context-aware entity representations embody meaningful semantics: entities with similar contexts are likely to live in proximity in the embedding space. Some other works extract and model contexts in an explicit manner with structured annotations. Structured annotations such as dependency parsing (Qu et al., 2017), user click information (Wei et al., 2009), or signed heterogeneous graphs (Ren & Cheng, 2015) are introduced to guide synonym discovery. 2) **Diverse Contexts**. An entity can be mentioned under a wide range of circumstances. Previous works on context-based synonym discovery either focus on entity information

only (Neculoiu et al., 2016; Mueller & Thyagarajan, 2016), or use a single piece of context for each entity (Liao et al., 2017; Qu et al., 2017) to learn a similarity function for entity matching. While in practice, similar context is only a sufficient but not necessary condition for context matching. Notably, in some domains such as medical, the context expression preference varies a lot from individuals. For example, `sinus congestion` is usually referred by medical professionals in the medical literature, while patients often use `stuffy nose` on social media. It is not practical to assume that each piece of context is equally informative to represent the meaning of an entity: a context may contribute differently when matched with different contexts of other entities. Thus it is imperative to focus on multiple pieces of contexts with a dynamic matching schema for accuracy and robustness.

In light of these challenges, we propose a framework to discover synonym entities from a massive corpus without additional structured annotation. Candidate entities are obtained from a massive text corpus unsupervisedly. A novel neural network model SYNONYMNET is proposed to detect entity synonyms based on two given entities via a bilateral matching among multiple pieces of contexts in which each entity appears. A leaky unit is designed to explicitly alleviate the noises from uninformative context during the matching process.

The contribution of this work is summarized as follows:

- We propose SYNONYMNET, a context-aware bilateral matching model to detect entity synonyms. SYNONYMNET utilizes multiple pieces of contexts in which each entity appears, and a bilateral matching schema with leaky units to determine entity synonymity.

- We introduce a synonym discovery framework that adopts SYNONYMNET to obtain synonym entities from a free-text corpus without additional structured annotation.

- Experiments on generic and domain-specific real-world datasets in English and Chinese demonstrate the effectiveness of the proposed model for synonym discovery.

## 2 SYNONYMNET

We introduce SYNONYMNET, our proposed model that detects whether or not two entities are synonyms to each other based on a bilateral matching between multiple pieces of contexts in which entities appear. Figure 1 gives an overview of the proposed model.

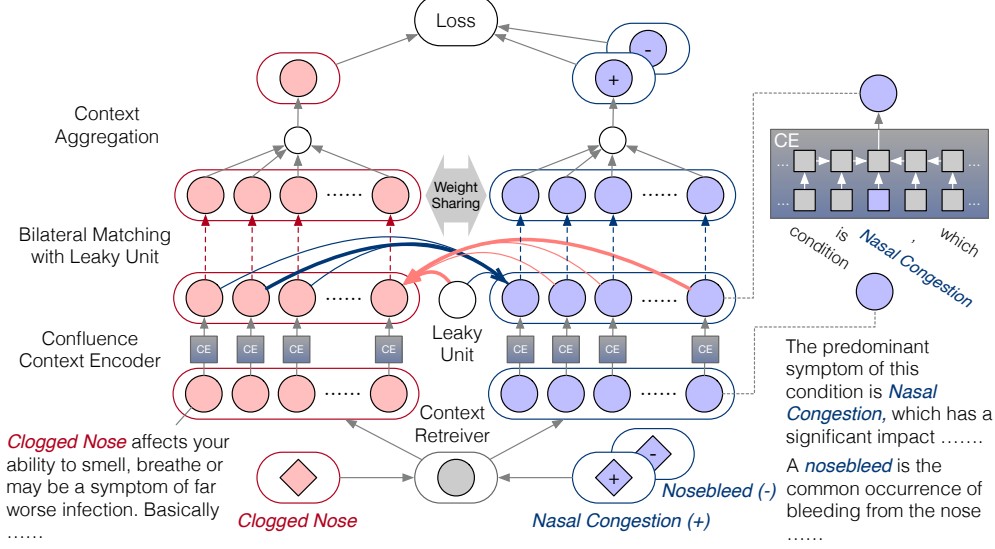

Figure 1: Overview of the proposed model SYNONYMNET. The diamonds are entities. Each circle is associated with a piece of context in which an entity appears. SYNONYMNET learns to minimize the loss calculated using multiple pieces of contexts via bilateral matching with leaky units.

## 2.1 CONTEXT RETRIEVER

For each entity $e$, the context retriever randomly fetches $P$ pieces of contexts from the corpus $D$ in which the entity appears. We denote the retrieved contexts for $e$ as a set $C = \{c_0, c_1, ..., c_P\}$, where $P$ is the number of context pieces. Each piece of context $c_p \in C$ contains a sequence of words $c_p = (w_p^{(0)}, w_p^{(1)}, ..., w_p^{(T)})$, where $T$ is the length of the context, which varies from one instance to another. $w_p^{(t)}$ is the $t$-th word in the $p$-th context retrieved for an entity $e$.

## 2.2 CONFLUENCE CONTEXT ENCODER

For the $p$-th context $c_p$, an encoder tries to learn a continuous vector that represents the context. For example, a recurrent neural network (RNN) such as a bidirectional LSTM (Bi-LSTM) (Hochreiter & Schmidhuber, 1997) can be applied to sequentially encode the context into hidden states:

$$\overrightarrow{\mathbf{h_p^{(t)}}} = \text{LSTM}_{fw}(\mathbf{w}_p^{(t)}, \overrightarrow{\mathbf{h_p^{(t-1)}}}), \qquad \overleftarrow{\mathbf{h_p^{(t)}}} = \text{LSTM}_{bw}(\mathbf{w}_p^{(t)}, \overleftarrow{\mathbf{h_p^{(t+1)}}}), \tag{1}$$

where $\mathbf{w}_p^{(t)}$ is the word embedding vector used for the word $w_p^{(t)}$. We could concatenate the last hidden state $\overrightarrow{\mathbf{h_p^{(T)}}}$ in the forward $\text{LSTM}_{fw}$ with the first hidden state $\overleftarrow{\mathbf{h_p^{(0)}}}$ from the backward $\text{LSTM}_{bw}$ to obtain the context vector $\mathbf{h}_p$ for $c_p$: $\mathbf{h}_p = [\overrightarrow{\mathbf{h_p^{(T)}}}, \overleftarrow{\mathbf{h_p^{(0)}}}]$. However, such approach does not explicitly consider the location where the entity is mentioned in the context. As the context becomes longer, it is getting risky to simply rely on the gate functions of LSTM to properly encode the context.

We introduce an encoder architecture that models contexts for synonym discovery, namely the confluence context encoder. The confluence context encoder learns to encode the local information around the entity from the raw context, without utilizing additional structured annotations. It focuses on both forward and backward directions. However, the encoding process for each direction ceases immediately after it goes beyond the entity word in the context: $\mathbf{h}_p = [\overrightarrow{\mathbf{h_p^{(t_e)}}}, \overleftarrow{\mathbf{h_p^{(t_e)}}}]$, where $t_e$ is the index of the entity word $e$ in the context and $\mathbf{h}_p \in \mathbb{R}^{1 \times d_{CE}}$. By doing this, the confluence context encoder summarizes the context while explicitly considers the entity's location in the context, where no additional computation cost is introduced.

Comparing with existing works for context modeling (Cambria et al., 2018) where the left context and right context are modeled separately, but with the entity word being discarded, the confluence context encoder preserves entity mention information as well as the inter-dependencies between the left and right contexts.

## 2.3 BILATERAL MATCHING WITH LEAKY UNIT

Considering the base case, where we want to identify whether or not two entities, say $e$ and $k$, are synonyms with each other, we propose to find the consensus information from multiple pieces of contexts via a bilateral matching schema. Recall that for entity $e$, $P$ pieces of contexts $H = \{\mathbf{h}_1, \mathbf{h}_2, ..., \mathbf{h}_P\}$ are randomly fetched and encoded. And for entity $k$, we denote $Q$ pieces of contexts being fetched and encoded as $G = \{\mathbf{g}_1, \mathbf{g}_2, ..., \mathbf{g}_Q\}$. Instead of focusing on a single piece of context to determine entity synonymity, we adopt a bilateral matching between multiple pieces of encoded contexts for both accuracy and robustness.

$H{\to}G$ matching phrase: For each $\mathbf{h_p}$ in $H$ and $\mathbf{g_q}$ in $G$, the matching score $m_{p \to q}$ is calculated as: $m_{p \to q} = \frac{\exp(\mathbf{h}_p \mathbf{W}_{\text{BM}} \mathbf{g}_q^{\text{T}})}{\sum\limits_{p' \in P} \exp(\mathbf{h}_{p'} \mathbf{W}_{\text{BM}} \mathbf{g}_q^{\text{T}})}$, where $\mathbf{W}_{\text{BM}} \in \mathbb{R}^{d_{CE} \times d_{CE}}$ is a bi-linear weight matrix.

Similarly, the $H{\leftarrow}G$ matching phrase considers how much each context $\mathbf{g}_q \in G$ could be useful to $\mathbf{h}_p \in H$: $m_{p \leftarrow q} = \frac{\exp(\mathbf{g}_q \mathbf{W}_{\text{BM}} \mathbf{h}_p^{\text{T}})}{\sum\limits_{q' \in Q} \exp(\mathbf{g}_{q'} \mathbf{W}_{\text{BM}} \mathbf{h}_p^{\text{T}})}$. Note that $P \times Q$ matching needs to be conducted in total for each entity pair. We write the equations for each $\mathbf{h}_p \in H$ and $\mathbf{g}_q \in G$ for clarity. Regarding the implementation, the bilateral matching can be easily written and effectively computed in a matrix form, where a matrix multiplication is used $\mathbf{H}\mathbf{W}_{\text{BM}}\mathbf{G}^T \in \mathbb{R}^{P \times Q}$ where $\mathbf{H} \in \mathbb{R}^{P \times D_{CE}}$ and $\mathbf{G} \in \mathbb{R}^{Q \times D_{CE}}$. The matching score matrix $\mathbf{M}$ can be obtained by taking softmax on the $\mathbf{H}\mathbf{W}_{\text{BM}}\mathbf{G}^T$ matrix over certain axis (over 0-axis for $\mathbf{M}_{p \to q}$, 1-axis for $\mathbf{M}_{p \leftarrow q}$).

Not all contexts are informative during the matching for two given entities. For example, some contexts may contain intricate contextual information even if they mention the entity explicitly. In this work, we introduce a leaky unit during the bilateral matching, so that uninformative contexts can be routed via the leaky unit rather than forced to be matched with any informative contexts. The leaky unit is a domain-dependent vector $\mathbf{l} \in \mathbb{R}^{1 \times d_{CE}}$ learned with the model. For simplicity, we keep $\mathbf{l}$ as a zero vector. If we use the $H \rightarrow G$ matching phrase as an example, the matching score from the leaky unit $\mathbf{l}$ to the $q$-th encoded context in $\mathbf{g}_q$ is:

$$m_{l \rightarrow q} = \frac{\exp(\mathbf{l}\mathbf{W}_{\text{BM}}\mathbf{g}_q^{\text{T}})}{\exp(\mathbf{l}\mathbf{W}_{\text{BM}}\mathbf{g}_q^{\text{T}}) + \sum\limits_{p' \in P} \exp(\mathbf{h}_{p'}\mathbf{W}_{\text{BM}}\mathbf{g}_q^{\text{T}})}. \tag{2}$$

Then, if there is any uninformative context in $H$, say the $\tilde{p}$-th encoded context, $\mathbf{h}_{\tilde{p}}$ will contribute less when matched with $\mathbf{g}_q$ due to the leaky effect: when $\mathbf{h}_{\tilde{p}}$ is less informative than the leaky unit $\mathbf{l}$.

$$m_{\tilde{p} \rightarrow q} = \frac{\exp(\mathbf{h}_{\tilde{p}}\mathbf{W}_{\text{BM}}\mathbf{g}_q^{\text{T}})}{\exp(\mathbf{l}\mathbf{W}_{\text{BM}}\mathbf{g}_q^{\text{T}}) + \sum\limits_{p' \in P} \exp(\mathbf{h}_{p'}\mathbf{W}_{\text{BM}}\mathbf{g}_q^{\text{T}})}. \tag{3}$$

## 2.4 Context Aggregation

The informativeness of a context for an entity should not be a fixed value: it heavily depends on the other entity and the other entity's contexts that we are comparing with. The bilateral matching scores indicate the matching among multiple pieces of encoded contexts for two entities. For each piece of encoded context, say $\mathbf{g}_q$ for the entity $k$, we use the highest matched score with its counterpart as the relative informativeness score of $\mathbf{g}_q$ to $k$, denote as $a_q = \max(m_{p \rightarrow q}|p \in P)$. Then, we aggregate multiple pieces of encoded contexts for each entity to a global context based on the relative informativeness scores:

$$\text{for entity } e: \quad \bar{\mathbf{h}} = \sum\nolimits_{p \in P} a_p \mathbf{h}_p, \quad \text{for entity } k: \quad \bar{\mathbf{g}} = \sum\nolimits_{q \in Q} a_q \mathbf{g}_q. \tag{4}$$

Note that due to the leaky effect, less informative contexts are not forced to be heavily involved during the aggregation: the leaky unit may be more competitive than contexts that are less informative, thus assigned with larger matching scores. However, as the leaky unit is not used for aggregation, scores on informative contexts become more salient during context aggregation.

## 2.5 Training Objectives

We introduce two architectures for training the SYNONYMNET: a siamese architecture and a triplet architecture.

**Siamese Architecture** The Siamese architecture takes two entities $e$ and $k$, along with their contexts $H$ and $G$ as the input. The following loss function $L_{\text{Siamese}}$ is used in training for the Siamese architecture:

$$L_{\text{Siamese}} = yL_+(e, k) + (1 - y)L_-(e, k), \tag{5}$$

where it contains losses for two cases: $L_+(e, k)$ when $e$ and $k$ are synonyms to each other ($y = 1$), and $L_-(e, k)$ when $e$ and $k$ are not ($y = 0$). Specifically, inspired by Neculoiu et al. (2016), we have

$$L_+(e, k) = \frac{1}{4}(1 - s(\bar{\mathbf{h}}, \bar{\mathbf{g}}))^2,$$
$$L_-(e, k) = max(s(\bar{\mathbf{h}}, \bar{\mathbf{g}}) - m, 0)^2, \tag{6}$$

where $s(\cdot)$ is a similarity function, e.g. cosine similarity, and $m$ is the margin value. $L_+(e, k)$ decreases monotonically as the similarity score becomes higher within the range of [-1,1]. $L_+(e, k) = 0$ when $s(\bar{\mathbf{h}}, \bar{\mathbf{g}}) = 1$. For $L_-(e, k)$, it remains zero when $s(\bar{\mathbf{h}}, \bar{\mathbf{g}})$ is smaller than a margin $m$. Otherwise $L_-(e, k)$ increases as $s(\bar{\mathbf{h}}, \bar{\mathbf{g}})$ becomes larger.

**Triplet Architecture** The Siamese loss makes the model assign rational pairs with absolute high scores and irrational ones with low scores, while the rationality of entity synonymity could be quite relative to the context. The triplet architecture learns a metric such that the global context $\bar{\mathbf{h}}$ of an entity $e$ is relatively closer to a global context $\bar{\mathbf{g}}_+$ of its synonym entity, say $k_+$, than it is to the global context $\bar{\mathbf{g}}_-$ of a negative example $\bar{\mathbf{g}}_-$ by some margin value $m$. The following loss function $L_{\text{Triplet}}$ is used in training for the Triplet architecture:

$$L_{\text{Triplet}} = \max(s(\bar{\mathbf{h}}, \bar{\mathbf{g}}_-) - s(\bar{\mathbf{h}}, \bar{\mathbf{g}}_+) + m, 0). \tag{7}$$

## 2.6 INFERENCE

The objective of the inference phase is to discover synonym entities for a given query entity from the corpus effectively. We utilize context-aware word representations to obtain candidate entities that narrow down the search space. The SYNONYMNET verifies entity synonymity by assigning a synonym score for two entities based on multiple pieces of contexts. The overall framework is described in Figure 2.

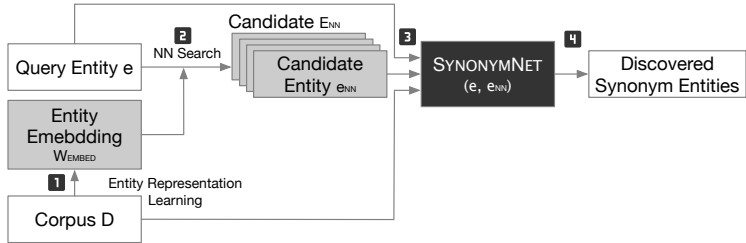

Figure 2: Synonym discovery during the inference phase with SYNONYMNET. (1): Obtain entity representations $\mathbf{W}_{\text{EMBED}}$ from the corpus $D$. (2): For each query entity $e$, search in the entity embedding space and construct a candidate entity set $E_{NN}$. (3): Retrieve contexts for the query entity $e$ and each candidate entity $e_{NN} \in E_{NN}$ from the corpus $D$, and feed the encoded contexts into SYNONYMNET. (4): Discover synonym entities of the given entity by the output of SYNONYMNET.

When given a query entity $e$, it is tedious and very ineffective to verify its synonymity with all the other possible entities. In the first step, we train entity representation unsupervisely from the massive corpus $D$ using methods such as skipgram (Mikolov et al., 2013) or GloVe (Pennington et al., 2014). An embedding matrix can be learned $\mathbf{W}_{\text{EMBED}} \in \mathbb{R}^{v \times d_{\text{EMBED}}}$, where $v$ is the number of unique tokens in $D$. Although these unsupervised methods utilize the context information to learn semantically meaningful representations for entities, they are not directly applicable to entity synonym discovery. However, they do serve as an effective way to obtain candidates as they tend to give entities with similar neighboring context words similar representations. For example, `nba championship`, `chicago black hawks` and `american league championship series` have similar representations because they tend to share some similar neighboring words. But they are not synonyms with each other.

In the second step, we construct a candidate entity list $E_{NN}$ by finding nearest neighbors of a query entity $e$ in the entity embedding space of $\mathbb{R}^{d_{\text{EMBED}}}$. Ranking entities by their proximities with the query entity on the entity embedding space significantly narrows down the search space for synonym discovery.

For each candidate entity $e_{NN} \in E_{NN}$ and the query entity $e$, we randomly fetch multiple pieces of contexts in which entities are mentioned, and feed them into the proposed SYNONYMNET model.

SYNONYMNET calculates a score $s(e, e_{NN})$ based on the bilateral matching with leaky units over multiple pieces of contexts. The candidate entity $e_{NN}$ is considered as a synonym to the query entity $e$ when it receives a higher score $s(e, e_{NN})$ than other non-synonym entities, or exceeds a specific threshold. In appendix A, we provide pseudo codes for the synonym discovery using SYNONYMNET.

## 3 EXPERIMENTS

### 3.1 EXPERIMENT SETUP

**Datasets** Three datasets are prepared to show the effectiveness of the proposed model on synonym discovery. The Wiki dataset contains 6.8M documents from Wikipedia[1] with generic synonym entities obtained from Freebase[2]. The PubMed is an English dataset where 0.82M research paper abstracts are collected from PubMed[3] and UMLS[4] contains existing entity synonym information in

---

[1] https://www.wikipedia.org/

[2] https://developers.google.com/freebase

[3] https://www.ncbi.nlm.nih.gov/pubmed

[4] https://www.nlm.nih.gov/research/umls/

the medical domain. The Wiki + FreeBase and PubMed + UMLS are public available datasets used in previous synonym discovery tasks (Qu et al., 2017). The MedBook is a Chinese dataset collected by authors where we collect 0.51M pieces of contexts from Chinese medical textbooks as well as online medical question answering forums. Synonym entities in the medical domain are obtained from MKG, a medical knowledge graph. Table 1 shows the dataset statistics.

| Dataset | Wiki + FreeBase | PubMed + UMLS | MedBooK + MKG |
|---|---|---|---|
| #ENTITY | 9274 | 6339 | 32,002 |
| #VALID | 394 | 386 | 661 |
| #TEST | 104 | 163 | 468 |
| #SYNSET | 4615 | 708 | 6600 |
| #CONTEXT | 6,839,331 | 815,644 | 514,226 |
| #VOCAB | 472,834 | 1,069,061 | 270,027 |

Table 1: Dataset Statistics.

**Preprocessing** Wiki +Freebase and PubMed + UMLS come with entities and synonym entity annotations, we adopt the Stanford CoreNLP package to do the tokenization. For MedBook, a Chinese word segmentation tool Jieba[5] is used to segment the corpus into meaningful entities and phrases. We remove redundant contexts in the corpus and filter out entities if they appear in the corpus less than five times. For entity representations, the proposed model works with various unsupervised word embedding methods. Here for simplicity, we adopt skip-gram (Mikolov et al., 2013) with a dimension of 200. Context window is set as 5 with a negative sampling of 5 words for training.

**Evaluation Metric** For synonym detection using SYNONYMNET and other alternatives, we train the models with existing synonym and randomly sampled entity pairs as negative samples. During testing, we also sample random entity pairs as negative samples to evaluate the performance. Note that all test synonym entities are from unobserved groups of synonym entities: none of the test entities is observed in the training data. Thus evaluations are done in a completely cold-start setting.

The area under the curve (AUC) and Mean Average Precision (MAP) are used to evaluate the model. AUC is used to measure how well the models assign high scores to synonym entities and low scores to non-synonym entities. An AUC of 1 indicates that there is a clear boundary between scores of synonym entities and non-synonym entities. Additionally, a single-tailed t-test is conducted to evaluate the significance of performance improvements when we compare the proposed SYNONYMNET model with all the other baselines.

For synonym discovery during the inference phase, we obtain candidate entities $E_{NN}$ from K-nearest neighbors of the query entity in the entity embedding space, and rerank them based on the output score $s(e, e_{NN})$ of the SYNONYMNET for each $e_{NN} \in E_{NN}$. We expect candidate entities in the top positions are more likely to be synonym with the query entity. We report the precision at position K (P@K), recall at position K (R@K), and F1 score at position K (F1@K).

**Baselines** We compare the proposed model with the following alternatives. (1) **word2vec** (Mikolov et al., 2013): a word embedding approach based on entity representations learned from the skip-gram algorithm. We use the learned word embedding to train a classifier for synonym discovery. A scoring function $Score_D(u, v) = x_u \mathbf{W} x_v^T$ is used as the objective. (2) **GloVe** (Pennington et al., 2014): another word embedding approach. The entity representations are learned based on the GloVe algorithm. The classifier is trained with the same scoring function $Score_D$, but with the learned glove embedding for synonym discovery. (3) **SRN** (Neculoiu et al., 2016): a character-level approach that uses a siamese multi-layer bi-directional recurrent neural networks to encode the entity as a sequence of characters. The hidden states are averaged to get an entity representation. Cosine similarity is used in the objective. (4) **MaLSTM** (Mueller & Thyagarajan, 2016): another character-level approach. We adopt MaLSTM by feeding the character-level sequence to the model. Unlike SRN that uses Bi-LSTM, MaLSTM uses a single direction LSTM and $l$-1 norm is used to measure the distance between two entities. (5) **DPE** (Qu et al., 2017): a model that utilizes dependency parsing results as the structured annotation on a single piece of context for synonym discovery. (6) **SYNONYMNET** is the proposed model, we used siamese loss (Eq. 6) and triplet loss (Eq. 7) as the objectives, respectively.

---

[5]https://github.com/fxsjy/jieba

## 3.2 PERFORMANCE EVALUATION

We first apply random search to obtain the best-performing hyperparameter setting on the validation set. The hyperparameter settings, as well as the sensitivity analysis, are reported in Appendix B. We report Area Under the Curve (AUC) and Mean Average Precision (MAP) on three datasets in Table 2. From the upper part of Table 2 we can see that SYNONYMNET performances consistently better than other baselines on three datasets. SYNONYMNET with the triplet training objective achieves the best performance on Wiki +Freebase, while the Siamese objective works better on PubMed +UMLS and MedBook + MKG. Word2vec is generally performing better than GloVe. SRNs achieve decent performance on PubMed + UMLS and MedBook + MKG. This is probably because the synonym entities obtained from the medical domain tend to share more character-level similarities, such as `6-aminohexanoic acid` and `aminocaproic acid`. However, even if the character-level features are not explicitly used in our model, our model still performances better, by exploiting multiple pieces of contexts effectively. DPE has the best performance among other baselines, by annotating each piece of context with dependency parsing results. However, the dependency parsing results could be error-prone for the synonym discovery task, especially when two entities share the similar usage but with different semantics, such as `NBA finals` and `NFL playoffs`. Additional results are reported in Appendix C.

We conduct statistical significance tests to validate the performance improvement. The single-tailed t-test is performed for all experiments, which measures whether or not the results from the proposed model are significantly better than ones from baselines. The numbers with † markers in Table 2 indicate that the improvement is significant with $p<0.05$.

| **MODEL** | **Wiki + Freebase** | | **PubMed + UMLS** | | **MedBook + MKG** | |
|---|---|---|---|---|---|---|
| | AUC | MAP | AUC | MAP | AUC | MAP |
| word2vec (Mikolov et al., 2013) | 0.9272 | 0.9371 | 0.9301 | 0.9422 | 0.9393 | 0.9418 |
| GloVe (Pennington et al., 2014) | 0.9188 | 0.9295 | 0.8890 | 0.8869 | 0.7250 | 0.7049 |
| SRN (Neculoiu et al., 2016) | 0.8864 | 0.9134 | 0.9517 | 0.9559 | 0.9419 | 0.9545 |
| MaLSTM (Mueller & Thyagarajan, 2016) | 0.9178 | 0.9413 | 0.8151 | 0.8554 | 0.8532 | 0.8833 |
| DPE (Qu et al., 2017) | 0.9461 | 0.9573 | 0.9513 | 0.9623 | 0.9479 | 0.9559 |
| **SYNONYMNET (Pairwise)** | 0.9831† | 0.9818† | **0.9838†** | **0.9872†** | **0.9685** | **0.9673** |
| w/o Leaky Unit | 0.9827† | 0.9817† | 0.9815† | 0.9847† | 0.9667 | 0.9651 |
| w/o Confluence Encoder (Bi-LSTM) | 0.9683† | 0.9625† | 0.9495 | 0.9456 | 0.9311 | 0.9156 |
| **SYNONYMNET (Triplet)** | **0.9877†** | **0.9892†** | 0.9788† | 0.9800† | 0.9410 | 0.9230 |
| w/o Leaky Unit | 0.9705† | 0.9631† | 0.9779† | 0.9821† | 0.9359 | 0.9214 |
| w/o Confluence Encoder (Bi-LSTM) | 0.9582† | 0.9531† | 0.9412 | 0.9288 | 0.9047 | 0.8867 |

Table 2: Test performance in AUC and MAP on three datasets. † indicates the significant improvement over all baselines ($p < 0.05$).

Besides numeric metrics, we also use box plots to represent the score distributions for each method on all three datasets in Figure 3. The red bars indicate scores on positive entity pairs that are synonym with each other, while the blue bars indicate scores on negative entity pairs. A general conclusion is that our model assigns higher scores for synonym entity pairs, marginally higher than other non-synonym entity pairs when compared with other alternatives.

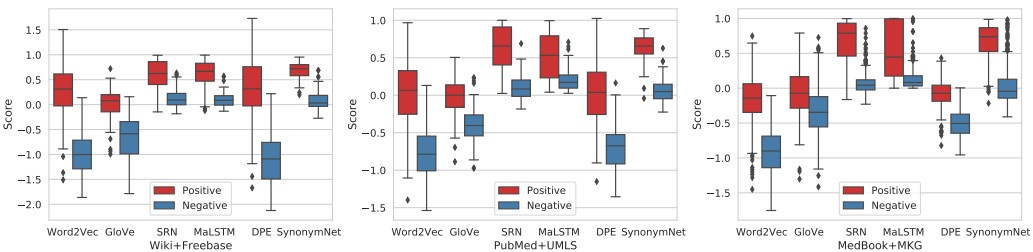

Figure 3: Test synonym score distributions on positive and negative entity pairs.

## 3.3 ABLATION STUDY

To study the contribution of different modules of SYNONYMNET for synonym discovery, we also report ablation test results in the lower part of Table 2. "w/o Confluence Context Encoder" uses the Bi-LSTM as the context encoder. The last hidden states in both forward and backward directions

in Bi-LSTM are concatenated; "w/o Leaky Unit" does not have the ability to ignore uninformative contexts during the bilateral matching process: all contexts retrieved based on the entity, whether informative or not, are utilized in bilateral matching. From the lower part of Table 2 we can see that both modules (Leaky Unit and Confluence Encoder) contribute to the effectiveness of the model. The leaky unit contributes 1.72% improvement in AUC and 2.61% improvement in MAP on the Wiki dataset when trained with the triplet objective. The Confluence Encoder gives the model an average of 3.17% improvement in AUC on all three datasets, and up to 5.17% improvement in MAP.

# 4 RELATED WORKS

## 4.1 SYNONYM DISCOVERY

The synonym discovery focuses on detecting entity synonyms. Most existing works try to achieve this goal by learning from structured information such as query logs (Ren & Cheng, 2015; Chaudhuri et al., 2009; Wei et al., 2009). While in this work, we focus on synonym discovery from free-text natural language contexts, which requires less annotation and is more challenging.

Some existing works try to detect entity synonyms by entity-level similarities (Lin et al., 2003; Roller et al., 2014; Neculoiu et al., 2016; Wieting et al., 2016). For example, Roller et al. (2014) introduce distributional features for hypernym detection. Neculoiu et al. (2016) use a Siamese structure that treats each entity as a sequence of characters, and uses a Bi-LSTM to encode the entity information. Such approach may be helpful for synonyms with similar spellings, or dealing with abbreviations. Without considering the context information, it is hard for the aforementioned methods to infer synonyms that share similar semantics but are not alike verbatim, such as `JD` and `law degree`.

Various approaches (Snow et al., 2005; Sun & Grishman, 2010; Liao et al., 2017; Cambria et al., 2018) are proposed to incorporate context information to characterize entity mentions. However, these models are not designed for synonym discovery. Qu et al. (2017) utilize additional structured annotations, e.g. dependency parsing result, as the context of the entity for synonym discovery. While we aim to discover synonym entities from a free-text corpus without structured annotation.

## 4.2 SENTENCE MATCHING

There is another related research area that studies sentence matching. Early works try to learn a meaningful single vector to represent the sentence (Tan et al., 2015; Mueller & Thyagarajan, 2016). These models do not consider the word-level interactions from two sentences during the matching. Wang & Jiang (2016); Wang et al. (2016; 2017) introduce multiple instances for matching with varying granularities. Although the above methods achieve decent performance on sentence-level matching, the sentence matching task is different from context modeling for synonym discovery in essence. Context matching focuses on local information, especially the words before and after the entity word; while the overall sentence could contain much more information, which is useful to represent the sentence-level semantics, but can be quite noisy for context modeling. We adopt a confluence encoder to model the context, which is able to aware of the location of an entity in the context while preserving information flow from both left and right contexts.

Moreover, sentence matching models do not explicitly deal with uninformative instances: max-pooling strategy and attention mechanism are introduced. The max-pooling strategy picks the most informative one and ignores all the other less informative ones. In context matching, such property could be unsatisfactory as an entity is usually associated with multiple contexts. We adopt a bilateral matching which involves a leaky unit to explicitly deal with uninformative contexts, so as to eliminate noisy contexts while preserving the expression diversity from multiple pieces of contexts.

# 5 CONCLUSIONS

In this paper, we propose a framework for synonym discovery from free-text corpus without structured annotation. A novel neural network model SYNONYMNET is introduced for synonym detection, which tries to determine whether or not two given entities are synonym with each other. SYNONYMNET makes use of multiple pieces of contexts in which each entity is mentioned, and compares the context-level similarity via a bilateral matching schema to determine synonymity. Experiments on three real-world datasets show that the proposed method SYNONYMNET has the ability to discover synonym entities effectively on both generic datasets (Wiki+Freebase in English), as well as domain-specific datasets (PubMed+UMLS in English and MedBook+MKG in Chinese) with an improvement up to 4.16% in AUC and 3.19% in MAP.

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

# APPENDIX

## A    PSEUDO CODE

Here we provide pseudo codes for the synonym discovery using SYNONYMNET.

**Data:** Candidate entity $e$, Entity Word Embeddings $W_{\text{EMBED}} \in \mathbb{R}^{v \times d}$, Document $D$
**Result:** Entity Set $K$ where each $k \in K$ is a synonym entity of $e$
$E_{NN}$ = NearestNeighbor($e$, $W_{\text{EMBED}}$)
Order $E_{NN}$ by the distance to $e$;
**for** $e_{NN}$ *in* $E_{NN}$ **do**
    Retrieve Contexts for $e_{NN}$ from Document $D$;
    Apply SYNONYMNET on $e$ and $e_{NN}$;
    **if** $s(e, e_{NN})$ >*threshold* **then**
        Add $e_{NN}$ as a synonym of $e$ to $K$;
    **end**
**end**

**Algorithm 1:** Effective Synonym Discovery via SYNONYMNET.

## B    HYPERPARAMETERS

We train the proposed model with a wide range of hyperparameter configurations, which are listed in Table 3. For the model architecture, we vary the number of randomly sampled contexts $P = Q$ for each entity from 1 to 20. Each piece of context is chunked by a maximum length of $T$. For the confluence context encoder, we vary the hidden dimension $d_{CE}$ from 8 to 1024. The margin value $m$ in triplet loss function is varied from 0.1 to 1.75. For the training, we try different optimizers (Adam (Kingma & Ba, 2014), RMSProp (Tieleman & Hinton, 2012), adadelta (Zeiler, 2012), and Adagrad (Duchi et al., 2011)), with the learning rate varying from 0.0003 to 0.01. Different batch sizes are used to train the model.

| HYPERPARAMETERS | VALUE |
|---|---|
| $P$ (context number) | {1, 3, 5, 10, 15, 20} |
| $T$ (maximum context length) | {10, 30, 50, 80} |
| $d_{CE}$ (layer size) | {8, 16, 32, 64, 128, 256, 512, 1024} |
| $m$ (margin) | {0.1, 0.25, 0.5, 0.75, 1.25, 1.5, 1.75} |
| Optimizer | {Adam, RMSProp, Adadelta, Adagrad} |
| Batch Size | {4, 8, 16, 32, 64, 128} |
| Learning Rate | {0.0003, 0.0001, 0.001, 0.01} |

Table 3: Hyperparameter settings.

| DATASETS | $P$ | $T$ | $d_{CE}$ | $m$ | Optimizer | Batch Size | Learning Rate |
|---|---|---|---|---|---|---|---|
| Wiki + Freebase | 20 | 50 | 256 | 0.75 | Adam | 16 | 0.0003 |
| PubMed + UMLS | 20 | 50 | 512 | 0.5 | Adam | 16 | 0.0003 |
| MedBook + MKG | 5 | 80 | 256 | 0.75 | Adam | 16 | 0.0001 |

Table 4: Hyperparaters.

Furthermore, we provide sensitivity analysis of the proposed model with different hyperparameters in Wiki + Freebase dataset in Figure 4. We first apply random search to obtain the best-performing hyperparameter setting on the validation dataset, as shown in Table 4. Figure 4 shows the performance curves when we vary one hyperparameter while keeping the remaining fixed. As the number of contexts $P$ increases, the model generally performs better. Due to limitations on computing resources, we are only able to verify the performance of up to 20 pieces of randomly sampled contexts.

The model achieves the best AUC and MAP when the maximum context length $T = 50$: longer contexts may introduce too much noise while shorter contexts may be less informative. A margin $m$ of 0.75 is used for training with triplet objective.

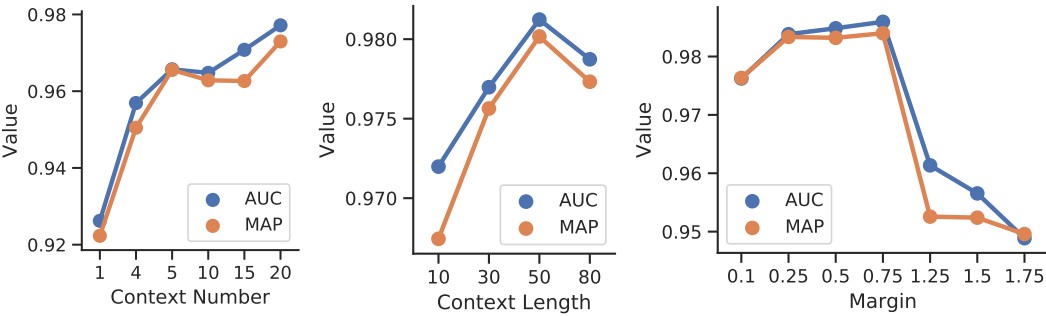

Figure 4: Sensitivity analysis.

## C  CASE STUDIES

We provide additional results and case studies in this section. Table 5 reports the performance in P@K, R@K, and F1@K. Table 6 shows a case for entity UNGA. The candidates are generated with pretrained word embedding using skip-gram.

| | Wiki + Freebase | | | PubMed + UMLS | | | MedBook + MedKG | | |
|---|---|---|---|---|---|---|---|---|---|
| | P@K | R@K | F1@K | P@K | R@K | F1@K | P@K | R@K | F1@K |
| K=1 | 0.3455 | 0.3455 | 0.3455 | 0.2400 | 0.0867 | 0.1253 | 0.3051 | 0.2294 | 0.2486 |
| K=5 | 0.1818 | 0.9091 | 0.3030 | 0.2880 | 0.7967 | 0.3949 | 0.2388 | 0.8735 | 0.3536 |
| K=10 | 0.1000 | 1.0000 | 0.1818 | 0.1800 | 1.0000 | 0.2915 | 0.1418 | 1.0000 | 0.2360 |

Table 5: Performance on Synonym Discovery.

| Candidate Entities | Cosine Similarity | Final Entities | SYNONYMNET Score |
|---|---|---|---|
| united_nations_general_assembly\|\|m.07vp7\|\| | 0.847374 | united_nations_general_assembly\|\|m.07vp7\|\| | 0.842602 |
| un_human_rights_council | 0.823727 | the_united_nations_general_assembly | 0.801745 |
| the_united_nations_general_assembly | 0.813736 | unga\|\|m.07vp7\|\| | 0.800719 |
| un_security_council\|\|m.07vnr\|\| | 0.794973 | | |
| palestine_national_council | 0.791135 | | |
| world_health_assembly\|\|m.05_gl9\|\| | 0.790837 | | |
| united_nations_security_council\|\|m.07vnr\|\| | 0.787999 | | |
| general_assembly_resolution | 0.784581 | | |
| the_un_security_council | 0.784280 | | |
| ctbt | 0.777627 | | |
| north_atlantic_council\|\|m.05pmgy\|\| | 0.775703 | | |
| resolution_1441 | 0.773064 | | |
| non-binding_resolution\|\|m.02pj22f\|\| | 0.771475 | | |
| unga\|\|m.07vp7\|\| | 0.770623 | | |

Table 6: Discovered Synonym Entities for UNGA using SYNONYMNET. A threshold of 0.8 on the SYNONYMNET score is used.

