# OpenReview forum: "SynonymNet: Multi-context Bilateral Matching for Entity Synonyms"
_ICLR.cc/2019/Conference_

### Official Review · AnonReviewer1 · 2018-10-29
**This paper presents a neural network model that detect synonymous entities based on contextual information without supervision.**

**Rating:** 4
**Confidence:** 4

**Review:**

Strengths:
- clear explanation of the problem
- clear explanation of the model and its application (pseudocode)
- clear explanation of training and resulting hyperparameters

Weaknesses:
- weak experimental settings:
-- (a) comparison against 'easy to beat' baselines. The comparison should also include as baselines the very relevant methods listed in the last paragraph of the related work section (Snow et a.l 2005, Sun and Grishman 2010, Liao et al. 2017, Cambria et al. 2018).
-- (b) unclear dataset selection: it is not clear which datasets are collected by the authors and which are pre-existing datasets that have been used in other work too. It is not clear if the datasets that are indeed collected by the authors are publicly available. Furthermore, no justification is given as to why well-known publicly available datasets for this task are not used (such as CoNLL-YAGO (Hoffart et al. 2011), ACE 2004 (NIST, 2004; Ratinov et al. 2011), ACE 2005 (NIST, 2005; Bentivogli et al. 2010), and Wikipedia (Ratinov et al. 2011)).
- the coverage of prior work ignores the relevant work of Gupta et al. 2017 EMNLP. This should also be included as a baseline.
- Section 2 criticises Mikolov et al.'s skip-gram model on the grounds that it introduces noisy entities because it ignores context structure. Yet, the skip-gram model is used in the preprocessing step (Section 3.1). This is contradictory and should be discussed.
- the definition of synonyms as entities that are interchangeable under certain contexts is well known and well understood and does not require a reference. If a reference is given, it should not be a generic Wikipedia URL.
- the first and second bulletpoint of contributions should be merged into one. They refer to the same thing.
- the paper is full of English mistakes. A proficient English speaker should correct them.

---

> ### Author Response · Authors · 2018-11-25
> **Response**
>
> We thank the reviewer for the comments and suggestions.
>
> For the mentioned related works, Snow et a.l 2005, Sun and Grishman 2010, Liao et al. 2017, Cambria et al. 2018, they are not designed for synonym discovery task, so we do not compare with them in the experiments. The mentioned related works introduce different ways to incorporate the context information when we are able to obtain external knowledge such as the entity ontologies, dependency parsing results. Snow et al. model the context by dependency path features extracted from parse trees. Their model aims to extract the hypernym (is-a) entity pairs from the sentence. Sun and Grishman use the dependency parsing results to devise an unsupervised model that clusters local contexts. The contexts are used to discover patterns expressing relationships between entities. Liao et al. propose to annotate entity mentions from the sentence using limited contexts in short search queries. Cambria et al. learn concept primitives for sentiment analysis. The model encodes the left context and right context separately while neglecting the target word for context modeling. A neural tensor layer is used to model the interactions between left/right context.
>
> The models mentioned above inspire us to devise the context encoder in SynonymNet that both 1) explicitly models the entity information using its contexts, and 2) does not use additional structured annotations for context modeling.
>
> For the datasets, we verify the performance of the proposed model on both generic and domain-specific datasets in English and Chinese. We updated the dataset descriptions in Section 3.1. Wiki+Freebase contains generic entities and their contexts from Wikipedia. The PubMed+UMLS and MedBook+MKG contain medical entities and their related context in medical literature. Both Wiki+Freebase and PubMed+UMLS are pre-existing English datasets that are publicly available and adopted in previous synonym discovery works [1][2]. The MedBooK+MKG is a Chinese dataset collected by the authors, which is complementary to the existing English datasets, and will be made publicly available. For other datasets, CoNLL-YAGO (Hoffart et al. 2011) lacks a large enough training set for our model. ACE 2004 (NIST, 2004; Ratinov et al. 2011) and ACE 2005 (NIST, 2005; Bentivogli et al. 2010) are less accessible due to copyright issues. The Wiki+Freebase dataset we used shares the same source with Wikipedia (Ratinov et al. 2011).
>
> For Gupta et al., this work mainly harnesses morphological regularities to deal with analogies like king – queen = man – woman. This is different from the studied task, i.e., synonym discovery.
>
> Thanks for the suggestions. We updated Section 2 and Section 3.1. Word embedding methods such as the skip-gram learn word representations effectively from a large-scale unannotated corpus. The semantics in the pre-trained embeddings make them suitable to initialize word embeddings for our model. The embeddings are updated as we train the model.
> Also, the learned word embeddings are suitable to search for candidate entities. Although the embeddings may involve noisy entities, they significantly narrow down the candidate searching space during inference phase: not all entities need to be verified with a target entity. The noisy candidates introduced by the initial word embeddings are further pruned away by the matching layer in the SynonymNet model.
>
> Thanks for your suggestions! We removed the reference for entities according to your suggestions for simplicity. To further clarify the technical novelty of the proposed algorithm, we have rephrased the contributions in Section 1.
>
> Thanks again, and the paper has been proofread for grammar errors.
>
> [1] Qu, Meng, Xiang Ren, and Jiawei Han. "Automatic synonym discovery with knowledge bases." Proceedings of the 23rd ACM SIGKDD International Conference on Knowledge Discovery and Data Mining. ACM, 2017.
> [2] https://github.com/mnqu/DPE

---

### Official Review · AnonReviewer2 · 2018-11-04
**Nice approach for automatically discovering synonymous entities**

**Rating:** 7
**Confidence:** 5

**Review:**

The paper presents a neural network model (SYNONYMNET) for automatically discovering synonymous entities from a large free-text corpus with minimal human annotation. The solution is fairly natural in the form of a siamese network, a class of neural network architectures that contain two or more identical subnetworks, which are an obvious approach for such a task, even though this task's SotA does not cover such architectures. even though the abstract consists the word novel, the chosen architecture is not a novel one but attached to this task, it can be considered as if.

# Paper discussion:

The introduction and the related work are well explained and the article is well structured. The authors mark very well the utility of automatically discovering synonyms.

Section 2 presents the SynonymNet, mainly the bi-LSTM applied on the contexts and the bilateral matching with leaky unit and the context aggregation for each entity, along with training objectives and the inference phase.

The novelty does not consist in the model since the model derives basically from a siamese network, but more in the approach, mainly the bilateral matching: one input is a context for an entity, the other input is a context for the synonym entity, and the output is the consensus information from multiple pieces of contexts via a bilateral matching schema with leaky unit (highest matched score with its counterpart as the relative informativeness score) and the context aggregation. The inference phase is a natural step afterward. Also, the usage of the leaky unit is clearly stated.

Section 3 presents the experimental phase, which is correct. The choice of LSTMs is understandable but other experiments could have been done in order to make clearer why it has been chosen. Regarding also the word embeddings choice, other experiments could have been completed (word2vec and GloVe have been competing with many other embeddings recently).

One noticed misspelling: GolVe (Pennington et al., 2014)

---

> ### Author Response · Authors · 2018-11-25
> **Response**
>
> We thank the reviewer for the appreciation and the supportive comments.

---

### Official Review · AnonReviewer3 · 2018-11-05
**Interesting paper though there is room for improvement**

**Rating:** 5
**Confidence:** 4

**Review:**

This paper studies the problem of identifying (discovering) synonymous entities. The paper proposes using the "contexts" of the entities as they occur in associated text corpora (e.g. Wiki) in the proposed neural-network based embedding approach for this task. The key novelties of the approach lie in the "matching" system used, where contexts of one entity are matched with that for the other entity to see how well they align with each other (which effectively determines the similarity of the two entities). Experiments are conducted on three different datasets to show the efficacy of the proposed approach.

Overall I found the paper to be an interesting read with some nice ideas mixed in. However I also had some concerns which are highlighted later down below, which I believe if addressed would lead to a very strong work.

Quality: Above average

In general the method seems to work somewhat better than the baselines and the method does have a couple of interesting ideas.

Clarity: Average

I found a few key details to be missing and also felt the paper could have been better written.

Originality: Average

The matching approach and use of the leaky units was interesting tidbits. Outside of that the work is largely about the application of such Siamese RNNs based networks to this specific problem. (The use of context of entities has already been looked at in previous works albeit in a slightly more limited manner)

Significance: Slightly below average

I am not entirely sold on the use of this approach for this problem given its complexity and unclear empirical gains vs more sophisticated baselines. The matching aspect may have some use in other problems but nothing immediately jumps out as an obvious application.

----

Strengths / Things I liked about the paper:

- In general the method is fairly intuitive and simple to follow which I liked.
- The matching approach was an interesting touch.
- Similarly for the "leaky" unit.
- Experiments conducted on multiple datasets.
- The results indicate improvements over the baselines considered on all the three datasets.

Weaknesses / Things that concerned me:

-  (W1) Slightly unfair baselines? One of the first things that struck me in the experimental results was how competitive word2vec by itself was across all three datasets. This made me wonder what would happen if we were to use a more powerful embedding approach say FastText, Elmo, Cove or the recently proposed BERT? (The proposed method itself uses bidirectional LSTMs)

Furthermore all of them are equally capable of capturing the contexts as well. An even more competitive (and fair) set of baselines could have taken the contexts as well and use their embeddings as well. Currently the word2vec baseline is only using the embedding of the entity (text), whereas the proposed approach is also provided the different contexts at inference time. The paper says using the semantic structure and the diverse contexts are weaknesses of approaches using the contexts, but I don't see any method that uses the context in an embedding manner -- say the Cove context vectors. If the claim is that they won't add any additional value above what is already captured by the entity it would be good to empirically demonstrate this.

- (W2) Significance testing: On the topic of experimentation, I was concerned that significance testing / error estimates weren't provided for the main emprical results. The performance gaps seem to be quite small and to me it is unclear how significant these gaps are. Given how important significance testing is as an empirical practice this seems like a notable oversight which I would urge the authors to address.

- (W3) Missing key details: There were some key aspects of the work that I thought were not detailed. Chief among these was the selection of the contexts for the entities. How was this? How were the 20 contexts identified? Some of these entities are likely far more common than just 20 sentences and hence I wonder how these were selected?

Another key aspect I did not see addressed: How were the entities identified in the text (to be able to find the contexts for them)? The paper claims that they would like to learn from minimal human annotations but I don't understand how these entity annotations in the text were obtained. This again seems like a notable oversight.

- (W4) Concerns about the method: I had two major concerns about the method:

(a) Complexity of method :  I don't see an analysis of the computational cost of the proposed method (which scales quadratically with P the number of contexts);

(b) Effect of redundant "informative" contexts: Imagine you have a number of highly informative contexts for an entity but they are all very similar to each other. Due to the way the matching scores are aggregated, these scores are made to sum to 1 and hence no individual score would be very high. Given that this is the final coefficient for the associated context, this seems like a significant issue right?

Unless the contexts are selected to be maximally diverse, it seems like this can essentially end up hurting an entity which occurs in similar contexts repeatedly. I would like to see have seen the rationale for this better explained.

(c) A smaller concern was understanding the reasoning behind the different loss functions in the siamese loss function with a different loss for the positive and the negative, one using a margin and one which doesn't. One which scales to 1/4, the other scaling to (1-m)^2. This seems pretty arbitrary and I'd like to understand this.

-(W5) Eval setting : My last concern was with the overall evaluation setup. Knowledge bases like Freebase are optimized for precision rather than recall, which is why "discovery" of new relations is important. However if you treat all missing relationships as negative examples then how exactly are you measuring the true ability of a method? Thus overall I'm pretty skeptical about all the given numbers simply because we know the KBs are incomplete, but are penalizing methods that may potentially discover relations not in the KB.

---

> ### Author Response · Authors · 2018-11-25
> **Response (Cont'd)**
>
> (W4) Thanks for the insights on these concerns.
> (a)	The complexity of method: the proposed model compares each piece of context of one entity h_p with each context g_q of another entity. Thus P*Q comparisons are needed for each entity pair. We write equations for each comparison for clarity. Regarding the implementations, the bilateral matching can be easily written in a matrix form, where a matrix multiplication is used H*W_BM*G^T, and the matching score matrix M can be obtained by taking softmax on the M matrix over certain axis (over 0-axis for M_{p->q}, 1-axis for M_{p<-q}). The context aggregation can also be done using a simple max-pooling operator. Thus, the proposed matching system is computational efficient via matrix multiplication, sum, softmax, and pooling. Moreover, the matching itself does not introduce additional model parameters except a domain-dependent context vector l and a bi-linear weight matrix W_BM.
> (b)	With redundant pieces removed, the contexts are randomly sampled, matched, and aggregated in the proposed matching system. As noisy contexts are prevalently observed, and contexts are randomly sampled, we propose to deal with uninformative contexts by the proposed bilateral matching with leaky units. How high-quality, complimentary, and informative contexts can be retrieved and collectively fused itself is an open and challenging research problem, which we would like to explore in-depth in our future works.
> (c)	The intuition behind such design is that when two entities are synonym with each other, we would like to have a low loss if a high similarity score is learned; when two entities are not synonym with each other, we would like to have a low loss when the similarity score is low. A similar loss function has been used in previous works such as in the SRN (Neculoiu et al., 2016) model. We updated descriptions with examples in Section 2.5.
>
> (W5) In the training for knowledge graph completion tasks where the objective aims to discover new entity pairs of certain relationships, classifiers are trained to determine the rationality of candidate entity pairs. It is routine to obtain corrupt correct triples (h, r, t) ∈ S by replacing entities, and construct incorrect triples as negative samples for training [3][4][5]. Experiment results show that this learning schema is effective on classification and link prediction tasks on knowledge graphs such as Freebase (Bollacker et al. 2008) and WordNet (Miller 1995). Similarly, we adopt this learning schema and obtain negative samples by replacing existing entities in synonym entity pairs with random ones on our synonym discovery task. There is a small probability that the randomly generated negative samples could be rational synonym pairs, but we found this learning schema effective, which is in accordance with the situations for knowledge graph completion tasks where precision and missing pairs also ubiquitously observed.
>
> [1] Peters, Matthew, et al. "Semi-supervised sequence tagging with bidirectional language models." Proceedings of the 55th Annual Meeting of the Association for Computational Linguistics (Volume 1: Long Papers). Vol. 1. 2017.
> [2] Peters, Matthew, et al. "Deep Contextualized Word Representations." Proceedings of the 2018 Conference of the North American Chapter of the Association for Computational Linguistics: Human Language Technologies, Volume 1 (Long Papers). Vol. 1. 2018.
> [3] Socher, Richard, et al. "Reasoning with neural tensor networks for knowledge base completion." Advances in neural information processing systems. 2013.
> [4] Wang, Zhen, et al. "Knowledge Graph Embedding by Translating on Hyperplanes." AAAI. Vol. 14. 2014.
> [5] Lin, Yankai, et al. "Learning entity and relation embeddings for knowledge graph completion." AAAI. Vol. 15. 2015.

---

> ### Author Response · Authors · 2018-11-25
> **Response**
>
> We thank the reviewer for the thorough review and constructive feedback.
>
> We first would like to thank the reviewer for the positive feedback on our work.
>
> For the part that concerned the reviewer, we elaborate point by point as shown below:
>
> (W1) For the experiment setting, the proposed model can work with various word embeddings. The contribution of our work does not lie in the choice of word embeddings, but the proposed architecture that utilizes entity representations for bilateral matching among multiple pieces of contexts. Our model is independent of the choice of word embeddings, and we adopt Word2vec as a base case. We aim to experiment the modeling ability of different model architectures given the same word representation information for synonym discovery. With sophisticated word embedding methods such as Elmo or BERT, which achieve decent performances on various NLP tasks, we do expect that both baselines and our model will get better performance.
>
> (W2) We’ve added the significance testing in the experiment and update Table 2 with discussions. A single-tailed t-test is performed to see whether or not the proposed model can outperform other baselines with significant improvements.
>
> (W3) For the missing key details, the contexts are randomly selected from all contexts in which each entity is mentioned. Due to limitations on computing resources, we are only able to verify the performance of up to 20 pieces of randomly chosen contexts in which each entity is mentioned. For Wiki+Freebase and PubMed+UMLS, the datasets come with entity mentions annotated. While in MedBook+MKG, we apply existing NER model [1] with contextualized embeddings [2] to obtain the annotated entities from the text. We clarified the claim about the annotation: the proposed model does not require additional structured annotations on the free-text corpus, such as entity ontologies, dependency parsing results during training and inference. The inference stage for synonym discovery is also designed to be data-driven so that we do not need pre-specified candidate entity pairs prepared by domain experts to be verified by the model, which further alleviates annotation efforts. We added these details in the revised version.

---

### Meta-Review · Area_Chair1 · 2018-12-14

**Confidence:** 4
**Recommendation:** Reject

**Metareview:**

This paper presents a model to identify entity mentions that are synonymous.  This could have utility in practical scenarios that handle entities.

The main criticism of the paper is regarding the baselines used.  Most of the baselines that are compared against are extremely simple.  There is a significant body of literature that models paraphrase and entailment and many of those baselines are missing (decomposable attention, DIIN, other cross-attention mechanisms).  Adding those experiments would make the experimental setup stronger.

There is a bit of a disagreement between reviewers, but I agree with the two reviewers who point out the weakness of the experimental setup, and fixing those issues could improve the paper significantly.